# WebDS: An End-to-End Benchmark for Web-based Data Science

**Ethan Hsu**[1*], **Hong Meng Yam**[1,2*], **Ines Bouissou**[3†], **Aaron Murali John**[3†],
**Raj Thota**[3], **Josh Koe**[3], **Vivek Sarath Putta**[3], **G K Dharesan**[4],
**Alexander Spangher**[1], **Shikhar Murty**[1], **Tenghao Huang**[5], **Christopher D. Manning**[1]

[1]Stanford University    [2]Pinetree Research    [3]University of California, Berkeley
[4]Singapore University of Technology and Design    [5]University of Southern California
{ethanhsu, hongmeng, manning}@stanford.edu

[*]Equal contribution (first authors). [†]Equal contribution (second authors).

## Abstract

Many real-world data science tasks involve complex web-based interactions: finding appropriate data available on the internet, synthesizing multimodal data from different locations, and producing summarized analyses. Existing *web benchmarks* often focus on simplistic interactions and often do not require diverse tool-using capabilities. Conversely, traditional *data science benchmarks* typically concentrate on static, highly structured datasets and do not assess end-to-end workflows that encompass data acquisition, cleaning, analysis, and insight generation. In response, we introduce WebDS, the first end-to-end web-based data science benchmark. It comprises 870 web-based data science tasks across 29 diverse websites from structured government data portals to unstructured news media, challenging agents to perform complex, multi-step, tool-based operations, across heterogeneous data formats, to better reflect the realities of modern data analytics. Evaluations of current SOTA LLM agents indicate significant performance gaps in accomplishing these tasks. For instance, Browser Use, which accomplishes 80% of tasks on WebVoyager, completes only 15% of tasks in WebDS, which our analysis suggests is due to new failure modes, such as poor information grounding, repetitive behavior and shortcut-taking that agents performing WebDS's tasks display. By contrast, humans achieve around 90% accuracy, highlighting a substantial gap between current agents and human performance. By providing a more robust and realistic testing ground, WebDS sets the stage for significant advances in the development of practically useful LLM-based data science.

The WebDS benchmark is available at `https://webdsbenchmark.github.io/`

## 1 Introduction

Large language models (LLMs) with web-traversal and tool-use abilities, i.e., *agents*, have shown promise in utilizing the web's vast repository of knowledge: navigating websites, synthesizing insights, and executing tasks with minimal human intervention[1] (Hong et al., 2024). These agents hold significant potential for use in data science applications too, with researchers using agents to perform autonomous data querying, visualization, and prediction (Peasley et al., 2025; Jansen et al., 2025). Yet, combining the two tasks to extract data-driven insights from websites – an essential routine for most data science practitioners (Kandel et al., 2012; Sun et al., 2022) – is not trivial. Modern webpages are complex and dynamic: unstructured, multimodal data are scattered across multiple layers of interactivity with varying levels of access restrictions. Autonomously *navigating* the web to find data, *synthesizing* and *analyzing* it repeatedly, requires robust reasoning capabilities and contextual understanding.

Prior benchmarks fail to capture the scope of this challenge. Evaluations are narrowly targeted either towards (a) *web browsing* only (Zhou et al., 2024; Pan et al., 2024b), or (b) *data science* tasks only. Existing *web agent benchmarks* tend to study agents' ability to follow natural language instructions to

---

[1]For example, see recent advances like Deep Research tools by OpenAI (https://openai.com/index/introducing-deep-research/) and Google DeepMind

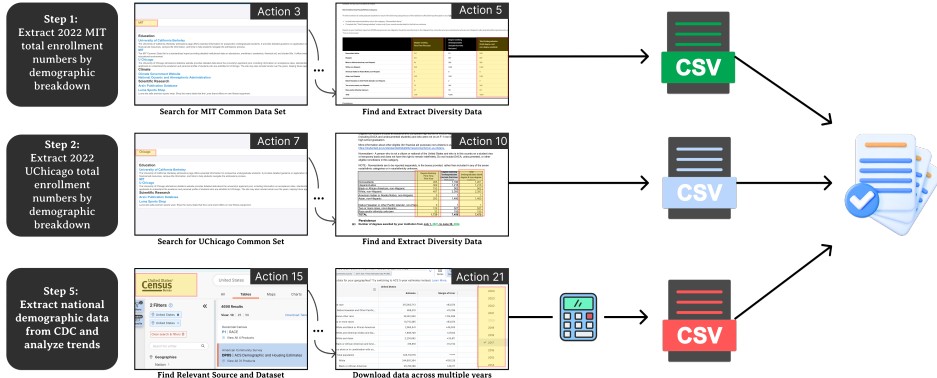

Figure 1: Example of the task, "Analyze the total enrollment numbers by racial/ethnic category for undergraduates (both degree- and non-degree-seeking) as of October 19, 2022. Cross-reference these numbers with national demographic trends and discuss the potential impact on the university's diversity initiatives. Write a report for the university's strategic planning committee on these trends and recommendations." *Note: task was selected to save space while still illustrating the multisite attribute.*

accomplish simple web-based tasks (e.g., writing Reddit posts or buying items (Zheng et al., 2024)), without assessing agents on their ability to manipulate and understand data. Additionally, these benchmarks often lack long-term robustness: WebVoyager (He et al., 2024), for instance, uses live sites that change over time, hindering reproducibility. Others (Li et al., 2020; Deng et al., 2024; Zhou et al., 2024) rely on coarse metrics that miss finer agent behaviors. On the other hand, existing *data science benchmarks* primarily emphasize data manipulation within code-based environments or structured spreadsheets (Li, 2025), or focus on querying unstructured data from static databases (Rajpurkar et al., 2016). However, realistic workflows of data scientists and other knowledge workers often begin by browsing the web, navigating across multiple websites, and synthesizing information from heterogeneous sources. Thus, browsing, searching and synthesizing are a fundamental, yet often overlooked, components of real-world web-based data science tasks Kandel et al. (2012); Sun et al. (2022).

To address the limitations of existing web agent and data science benchmarks, we introduce a novel benchmark, WebDS. WebDS consists of 29 data-rich websites (e.g., CDC, sales) and contains 870 real, human-written tasks that demand complex navigation and analysis. These tasks pose a challenge for leading models: while GPT-4o with BrowserUse obtains an $81.1\%$ success rate on WebVoyager (Müller & Žunič, 2024) it only obtains $13.2\%$ on WebDS; while GPT-4o with AgentOccam (Yang et al., 2024)[2] achieves a $45.7\%$ success rate on WebArena, it only obtains $4.8\%$ on WebDS. Surprisingly, we find that increasing model capacity does *not* necessarily lead to increased performance: GPT-4o performs similarly to GPT-4o-mini and Qwen2.5-72b. An error analysis (Section 7) shows significant, novel failure modes for all models: poor information grounding (where grounded knowledge contradicts latent knowledge), repetitive behavior and shortcut-taking (due to the multi-hop nature of web-based data science). Additionally, end-to-end model performance on WebDS is markedly below human performance: while the strongest agent reaches 13.2% success, a human baseline under identical constraints achieves 90% ($\pm 3\%$). This quantifies a large human–agent gap on realistic, long-horizon workflows.

In sum, WebDS introduces a significantly more challenging, real-world benchmark that represents a next frontier for model development. By bridging the gap between web interaction and data science capabilities, we hope that our work guides the community and pushes us towards developing novel end-to-end AI skills.

Our contributions are threefold:

1. **Comprehensive Task Suite** We introduce WebDS, a large-scale benchmark consisting of 870 tasks across 29 data-rich websites spanning 10 domains (discussed in section 3 and 4). The benchmark is released in two tracks: WebDS-live, which evaluates agents on live websites, and WebDS-dockerized, which provides a reproducible subset of these environments through containerized deployments and is the most comprehensive dockerized benchmark to date. These websites span more data

---

[2]This was the SOTA open source model on WebArena as of 5/15/2025.

types, modalities, and domains than any prior benchmark and these tasks require not only analytical reasoning but also interaction with diverse tools and interfaces.

2. **Realistic End-to-End Evaluation** WebDS is the first benchmark to assess execution of the full data science pipeline, simulating end-to-end workflows. Tasks begin with autonomous web browsing for relevant data, followed by analysis and/or visualization, and culminate in generating well-reasoned, context-aware outputs. A key contribution is that we quantify the large human–agent gap on these tasks, highlighting a unique and previously under-measured limitation that today's agents cannot yet sustain coherent performance across full data-science workflows.

3. **Reproducible, Realistic and Fine-Grained Evaluation.** WebDS combines the strengths of both reproducible and real-world evaluation paradigms. The WebDS-live track captures the evolving nature of real websites and reflects the challenges faced by practical web-based data science agents. In parallel, WebDS-dockerized provides containerized website environments whose state is preserved, enabling fully reproducible experiments and stable longitudinal benchmarking. Beyond prior binary metrics, WebDS introduces fine-grained measures of subtask completion, tool use, reasoning quality, and report fidelity, enabling both holistic evaluation and precise diagnostic analysis of agent performance.

## 2 RELATED WORK

**Data Analysis Benchmarks** Benchmarks such as SQuAD (Rajpurkar et al., 2016) and HotpotQA (Yang et al., 2018) have played critical roles in evaluating language models' reasoning over structured and semi-structured data, providing passage-based and multi-hop question-answering tasks, respectively. Further, Wikitable (Pasupat & Liang, 2015) offers a challenge in executing compositional logical queries over semi-structured tables, testing models' ability to generalize across complex tabular reasoning tasks. Recently, InfiAgent-DABench (Hu et al., 2024) emerged as the first specialized benchmark for data analysis agents, comprising 311 tasks across 55 CSV files with automated evaluation. Additional data science agent benchmarks study broader workflows, including DSBench (Jing et al., 2024) which evaluates data science tasks with Kaggle and modeloff datasets, DA-Code (Huang et al., 2024) which focuses on agentic code generation for data analysis, Spider 2.0 (Lei et al., 2024) which targets real enterprise text to SQL pipelines, Spider2-V (Cao et al., 2024) which extends these workflows into multimodal settings, and DABStep (Egg et al., 2025) which benchmarks multi step data reasoning abilities of LLMs. However, these benchmarks' focus on structured datasets limits assessment of real-world iterative workflows: data science workflows often heavily involve finding data (e.g., through web navigation), using tools, and analyzing unstructured data (Li, 2025; Rajpurkar et al., 2016).

**Web Agent Benchmarks** Evaluating LLM agents on high-level natural language tasks has traditionally relied on surface-form comparisons of action sequences, as seen in VirtualHome (Puig et al., 2018) and Mind2Web (Deng et al., 2024), but these fail to capture functional correctness. WebArena improves on this by evaluating functional correctness but has major limitations, including a lack of scoring granularity and rigid state-based evaluation that is not robust to minor variations (Zhou et al., 2024). Recent approaches use LLMs like GPT-4o to assess task completion (Pan et al., 2024a). However, these works rely only on final screenshots, making the evaluation binary and incomplete. Lastly, methods like WebVoyager consider the last 15 screenshots, greatly strengthening evaluation, but only gives a binary 'SUCCESSFUL / UNSUCCESSFUL' result, and is unable to account for full trajectory in complex tasks (He et al., 2024). Our work avoids this limitation by assessing the full trajectory, and allowing for granular analysis, which is especially important for our tasks which are multihop, multisite and generally involve more steps.

## 3 BENCHMARK DESIGN

In this section, we describe WebDS, the first web-based benchmark to focus on the full data science workflow. Our main objective is to produce a benchmark that is *useful*, *comprehensive*, *reproducible*, and *granular*.

**Subject Interviews** To ground the design of our benchmark and formulate task definitions, we conducted 8 interviews with journalists, data scientists, and other domain experts to identify recurring data-related needs. From these interviews, we derived two broad categories of tasks: those that (1) *culminate in a downstream product* and (2) those that *address a key analytical question*. Within these categories, tasks could involve a variety of data modalities and attributes, including: structured data,

| Dataset | Multihop | Structured | Unstructured | Web Nav | QA | Multisite | Actions | Tool-Use |
|---|---|---|---|---|---|---|---|---|
| SQuAD (Rajpurkar et al., 2016) | ✗ | ✗ | ✓ | ✗ | ✗ | ✗ | ✗ | ✗ |
| Wikitable (Pasupat & Liang, 2015) | ✓ | ✓ | ✗ | ✗ | ✗ | ✗ | ✗ | ✗ |
| HotpotQA (Yang et al., 2018) | ✓ | ✗ | ✓ | ✗ | ✗ | ✗ | ✗ | ✗ |
| WebVoyager (He et al., 2024) | ✗ | ✗ | ✗ | ✓ | ✓ | ✗ | ✓ | ✗ |
| WebArena (Zhou et al., 2024) | ✓ | ✗ | ✗ | ✓ | ✓ | ✓ | ✓ | ✓ |
| GAIA (Mialon et al., 2023) | ✓ | ✗ | ✓ | ✓ | ✓ | ✓ | ✗ | ✗ |
| WebWalker (Wu et al., 2025) | ✓ | ✗ | ✓ | ✓ | ✓ | ✓ | ✗ | ✗ |
| AssistantBench (Yoran et al., 2024) | ✓ | ✗ | ✓ | ✓ | ✓ | ✗ | ✓ | ✓ |
| **WebDS (ours)** | ✓ | ✓ | ✓ | ✓ | ✓ | ✓ | ✓ | ✓ |

Table 1: Comparison of dataset features across benchmarks. WebDS is the only benchmark that has tasks spanning Multihop, Structured, Unstructured, Web Nav, QA, Multisite. Actions, and Tool-Use.

unstructured textual and non-textual data, multi-hop reasoning, and multi-website data integration, all of which are discussed more in section 4. In light of these goals we have selected a set of **29** websites that cover **10** high-stakes data-heavy domains, and wrote by hand 870 tasks for LLM agents in these domains.

**Domains** We chose 29 websites covering 10 high-stakes domains, as shown in Table 7. In particular, we selected websites that included various diverse data formats, including table-structured data and downloadable CSVs as well as unstructured data such as text and graphics. Our selection criteria were: (1) websites had to be public, (2) they had to contain data and be used commonly in different kinds of data analysis and (3) they had to have a unique representation of data not found in the rest of our benchmark, or only found in one other website. For these reasons, high scores on our benchmark require an agent to be adept at reasoning and tool usage between different modalities and data representations.

**Action Space** We do not fix the action space $\mathcal{A}$, as widely adopted abstractions already exist (e.g., BrowserGym (Drouin et al., 2024), WebArena). This design allows flexibility: researchers may enable or disable tools (Python, Bash, SQL, etc.) without adapter friction.

**State and Observation Space** The state space consists of website pages and data accessible to the agent, with each state's observation consisting of HTML, DOM, screenshots, and/or AXTrees. Data can be downloaded, and many tasks necessitate the successful downloading and manipulation of data to complete. Each task begins at a fixed entry URL with a natural-language goal, and agents must autonomously discover relevant pages using their own navigation policies. We intentionally do not constrain crawling strategies or actions within each sandboxed-site, enabling users to plug in agents from other frameworks easily and improve compatibility with prior work.

**Reproducibility and Realism** Our benchmark supports two complementary evaluation regimes. In WebDS-live, agents interact directly with live websites. This track captures the full complexity of real web environments, including evolving page layouts and continually updated datasets, but may suffer from websites changing. To maintain evaluation stability, we record complete interaction trajectories, including page states, downloaded artifacts, and agent actions, allowing runs to be audited and analyzed post-hoc.
In parallel, WebDS-dockerized provides containerized deployments of a subset of benchmark websites whose content and structure are preserved at the time of benchmark construction. By freezing these environments within Docker containers, we enable deterministic execution and exact reproducibility of experiments. This dual-track design allows WebDS to balance realism with scientific rigor.

**Evaluation Granularity** Although prior benchmarks have provided some granularity in evaluating AI agent performance, we are the first to provide evaluation scores along three different dimensions: (1) *Domain-wise*, along different professional specializations (e.g., performance on demographics-related websites, sports-related websites, etc.) (2) *Attribute-wise*, along different types of challenges for LLM agents which we call "task attributes" (e.g., multihop, action-based, etc. see more in section 4) (3) *Difficulty*, along easy, medium, or hard task complexity.

**Release Strategy and Benchmark Integrity** To mitigate overfitting while preserving reproducibility, we release a 470-task public validation set and hold out a 400-task private test set. Leaderboards and official evaluations use rotating subsets of the private test set, where we publish prompts and accept predictions (JSON) for scoring on Huggingface. We may periodically refresh/expand the private test pool.

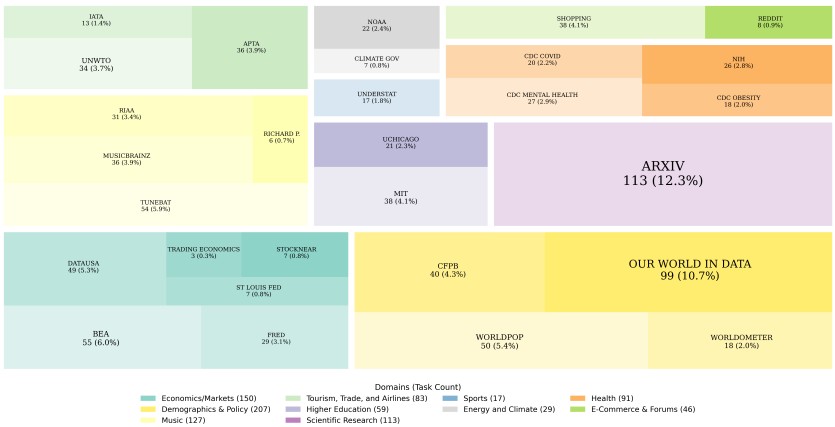

Figure 2: Distribution of tasks with respect to domain (rounded to nearest decimal), where each domain is a grouping of websites according to Table 7. Our domain distribution is chosen based on subject interviews (Section 3).

## 4 BENCHMARK TASKS

### 4.1 TASK FORMULATION

We define a web data science task as: *"a query which requires navigation within a browser-like environment to learn raw information that is then transformed to a predictive or summarized representation"*. Optionally, there can be a downstream action based on the resulting predictive representation. More formally, let: $\mathcal{W}$ denote the environment (e.g., the space of websites or a dynamic web environment), $\mathcal{Q}$ denote the space of data science queries, $\mathcal{D}$ denote the raw data space extracted from $\mathcal{W}$, $\mathcal{Y}$ denote the space of analytical outcomes (e.g., reports, models, visualizations), and $\mathcal{A}$ denote the set of downstream actions, with $\varnothing \in \mathcal{A}$ indicating no action. We define a web data science task as a composite mapping $f : \mathcal{W} \times \mathcal{Q} \to \mathcal{Y} \times \mathcal{A}$, where an agent, given a browser environment $\mathcal{W}$ and a natural language query $q \in \mathcal{Q}$, returns an analytical outcome $y \in \mathcal{Y}$ and/or an optional downstream action $a \in \mathcal{A}$. This overall process decomposes into three stages—information gathering, data analysis, and optional action—captured by the composition $f = f_\alpha \circ f_a \circ f_d$. In the first stage, $f_d : \mathcal{W} \times \mathcal{Q} \to \mathcal{D}$, the agent interacts with the environment to extract relevant raw data. In the second stage, $f_a : \mathcal{D} \to \mathcal{Y}$, the agent transforms this data into an analytical outcome, potentially through internal steps such as representation learning, tool usage, and prediction. In the final stage, $f_\alpha : \mathcal{Y} \to \mathcal{A}$, the agent maps the outcome to a downstream action—or none at all ($a = \varnothing$).

### 4.2 TASK GENERATION

To generate data science tasks that meet our formal definition, given in Section 4.1, we first identified 100 candidate websites that were freely accessible. Then, we selected a final set of 29 websites to ensure broad coverage of data modalities, enable the construction of multisite tasks, and represent a diverse range of task attributes. Finally, we grouped these websites into domains with similar content and structure, as shown in Figure 2. Using this curated set of websites, we wrote a total of 870 tasks, a simple example of which can be seen in Figure 1.

Tasks were authored by 8 primary annotators (paper authors). Every task underwent at least one secondary review for (i) solvability in the live site (at time of publication) and containerized deployment (if site is containerized), (ii) clarity of instructions, and (iii) correctness of references/ground truths; lead authors conducted a final pass for consistency and coverage.

In authoring the tasks, we aimed to achieve a balanced distribution of difficulty levels, comprehensive coverage of the identified attributes, and strict adherence to our task definition. Task writing emphasized diversity, the inclusion of verifiable ground truths, and the ability to evaluate a wide spectrum of real-world data science capabilities. We find that there is substantial diversity in task difficulty: some tasks can be completed through straightforward web searches, while others require synthesizing information from heterogeneous sources and applying a range of statistical or analytical techniques.

| | Task Category | Description / Example | Properties |
|---|---|---|---|
| QA | Single-hop QA (344 tasks) | *"What is the lowest AAPL stock price from 2000–2020?"* | - Single data source
- Question-Answering
- Webnav (likely)
- Possibly structured data
- No complex tool usage |
| | Multi-hop QA (117 tasks) | *"For each person $i$, let $X_i$ be the number of times their name appears in New York Times articles between [start date] and [end date]. Find the average and standard deviation of $X_i$ for all people mentioned."* | - Multiple data sources (articles)
- Question-Answering
- Unstructured data
- Tool usage (e.g., Python)
- Web navigation |
| Actions | Single-hop Action (97 tasks) | *"Delete the review from the user 'scammer Yoke' on our product page."* | - Single data source or site
- Requires an action (delete)
- Web navigation |
| | Multi-hop Action (134 tasks) | *"Buy the $n$th cheapest laptop after comparing prices across Store A, Store B, and Store C."* | - Multiple data sources (stores)
- Requires an action (purchase)
- Multi-hop + multi-website
- Web navigation |
| | Action + Tool Usage (139 tasks) | *"Compute the average number of people who got sick per [interval] using CDC data (csv file) and post the result on Reddit."* | - Structured data (csv)
- Tool usage (DB query, analytics)
- Action (post on Reddit)
- Web navigation |

Table 2: Illustrative examples of QA vs. Action-Based tasks, showing various properties (single- vs. multi-hop, structured vs. unstructured data, web navigation, and tool usage).

## 4.3 TASK ATTRIBUTES

Our tasks mirror real-world scenarios where a data analyst or general-purpose assistant must retrieve or compute information (which we call Question-Answering tasks) or perform downstream actions (which we call action-based tasks). Some tasks remain straightforward, requiring only a single data source or minimal steps, while others demand multi-hop reasoning, multi-website browsing, structured or unstructured data handling, and the integration of external tools.

We manually label each task with one or more of the following 7 labels, or "task attributes":

• **Question-Answering (QA)** Tasks where the system provides a definitive answer to a user query (e.g., compute the combined total assets for two funds). These tasks have *verifiable ground truths*.

• **Action-Based** Tasks that require the system to perform one or more actions after gathering and analyzing information. An example would be analyzing multiple data sources to rank financial indices and then posting an investment recommendation on Reddit.

• **Single-hop vs. Multi-hop** *Single-hop* tasks rely on a single data source to find the answer, while *multi-hop* tasks require combining information from multiple sources in the same website.

• **Structured vs. Unstructured (textual) vs. Unstructured (contextual) Tasks**. Structured tasks utilize data organized in a predefined format (such as databases). *Unstructured (textual)* tasks involve raw text data (e.g., documents or free-form responses), where extracting meaning requires natural language processing. *Unstructured (other)* tasks involve data *without* a clear textual representation. These tasks require interpretation of external knowledge, images or other multi-modal inputs.

• **Tool Usage** Some tasks necessitate the use of external tools (like Python scripts, Wolfram Alpha, or SQL queries) to perform computations or process information.

• **Web Navigation (Webnav)** Tasks that involve interacting with websites—whether querying a database online or making a post on a forum—relying on navigating through web pages.

• **Multi-website Tasks** Tasks that require gathering information or carrying out actions across multiple websites, where each site contributes separate pieces of the required data or functionality.

We then use these attributes to categorize tasks into three levels of difficulty: easy, medium and hard, based on specific attributes that reflect increasing cognitive and operational complexity, as per Table 3. See Table 2 for a more detailed breakdown of task-categories, but note that because these attributes are multi-label, some attributes (e.g., "Tool Usage") are in multiple categories. See Appendix A, Table 4 for more complete counts of tasks.

| Difficulty | Task Attributes |
|---|---|
| **Easy (247 tasks)** | Does not involve any of the following properties: multihop, nontextual unstructured data, action-based tasks, or tool-use and is single-website. |
| **Medium (275 tasks)** | Involves exactly one of the above properties and is single-website. |
| **Hard (348 tasks)** | Involves at least two of the above properties **or** is multi-website. |

Table 3: Task difficulty categorization based on structural and content properties. We find that agents on average perform 2.5x better on easy tasks compared to medium or hard

| Metric | 5 LLM Judge Runs |
|---|---|
| Perfect status agreement | 96.3% |
| Perfect score agreement | 91.3% |
| Low variance ($0 < \text{stdev} \leq 1$) | 6.1% |
| High variance ($\text{stdev} > 1$) | 2.6% |

Table 4: Stability results across five runs of the evaluation harness.

# 5 EXPERIMENTAL SETUP

## 5.1 EVALUATION PROTOCOL

Our evaluation framework combines automated and subjective assessment to robustly evaluate agents in complex, multi-step tasks. In particular, we provide two evaluations for each task, an automated binary evaluation and a subjective 1–5 score given by an LLM judge.

**Automated Evaluation**  For tasks with reference ground truths (e.g., extracting factual statistics) we assign binary labels: *SUCCESSFUL / UNSUCCESSFUL*. These are determined by a language model comparing the agent's final output or action to ground truth.

**Subjective Evaluation via LLM**  To evaluate open-ended tasks (e.g., generating a Reddit post with statistics), we build on the *LLM-as-a-Judge* paradigm introduced by WebVoyager (He et al., 2024), which classified outcomes based solely on the final 15 screenshots of a trajectory. We extend this method in two key ways (1) we have the LLM provide a more informative 1–5 integer score, along with rationale and failure analysis, enabling more granular agent iteration; and (2) instead of relying on the final states alone, we evaluate the full trajectory by summarizing each observation and analyzing (observation, action, next observation) triplets throughout the task, resulting in more reliable and interpretable scores, which are assigned as follows:

$$R_{\text{llm}}(s_T) = \begin{cases} 5 & \text{fully satisfies } G \\ 4 & \text{mostly satisfies } G \\ 3 & \text{partially satisfies } G \\ 2 & \text{major errors, some progress} \\ 1 & \text{major errors, no progress} \end{cases}$$

**Human Validation of Evaluation Harness**  To validate our subjective scoring method, we first conducted a human verification study. We had CS undergraduate annotators independently rate a random sample of 400 task-trajectory pairs and found that our evaluation harness achieved 93% status agreement with the human assessments.

**Stability Validation of Evaluation Harness**  Secondly, we ran our scoring harness five times across 800 tasks for the best performing BrowserUse agent. To verify the robustness of the harness, we computed both status agreement (SUCCESS vs. NOT SUCCESS) and score agreement across 5 runs. For score agreement, if all 5 runs are not in perfect agreement, we look at the standard deviation of the scores across all 5 runs, with low variance classified as the scores being generally within +-1 point of each other (standard deviation of the 5 scores being less than 1). Results are summarized in Table 4. The standard deviation of per-run average LLM scores was 0.005, indicating very high stability and almost no variability.

## 5.2 MODELS

We benchmarked nine state-of-the-art (SOTA) web agents to evaluate performance on our newly proposed benchmark. To ensure consistency with prior work, we first evaluated the general-purpose GPT-4o and GPT-4o-mini agents using accessibility trees as observations, following the exact implementation described in *WebArena* (Zhou et al., 2024). Using the same observation space and implementation pipeline, we additionally benchmarked Claude Sonnet-3.5, Claude Haiku-3.5,

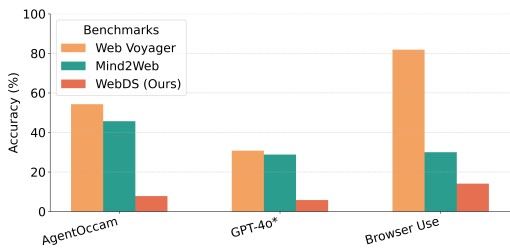

Figure 3: Agent Performance on Various Benchmarks. Models perform much worse on our benchmark (e.g., BrowserUse + GPT-4o achieves 81.9% on WebVoyager but 12.9% on WebDS). *Note: WebVoyager Performance is based on GPT-4, as no verified GPT-4o results exist (He et al., 2024).*

Qwen2.5-72B-Instruct, Claude Sonnet-4.5, GPT-5.1. To explore multimodal capabilities, we modified the observation space to incorporate full-page screenshots and evaluated Qwen2.5-VL-72B-Instruct under this setting.

We also included AgentOccam, the SOTA[3] open-source agent on *WebArena*, benchmarking it under the official configuration used for WebArena without altering its implementation. Finally, we evaluated BrowserUse (Müller & Žunič, 2024), the SOTA[3] agent on *WebVoyager* via both Qwen2.5-VL-72B-Instruct and GPT-4o. We made minimal adaptations to the observation and action space to better accommodate the structure of our landing pages and to improve handling of CSV files and file downloads.

### 5.3 HUMAN PERFORMANCE ON BENCHMARK

We recruited six participants with prior data-science experience. Participants were given the same browsing environment, websites, and tools as the evaluated agents, and were allotted at most 30 minutes per task. Note that the participants were using the live sites, which at the time of experiment were exactly identical to dockerized sites. All participants were familiarized with the interface and permitted one warm-up task for acclimatization before evaluation. Each was then assigned one of two disjoint, randomized subsets of benchmark tasks with reference ground truths. During evaluation, participants followed the same protocol as agents: starting from the specified URL, completing all navigation, data collection, and analysis steps directly in the browser environment.

## 6 RESULTS

| Agent | Success Rate (SR%) | | | | Score (1–5) | | | |
|---|---|---|---|---|---|---|---|---|
| | Overall | Easy | Medium | Hard | Overall | Easy | Medium | Hard |
| GPT-4o | 1.0 | 1.7 | 0.4 | 0.9 | 1.111 | 1.187 | 1.073 | 1.087 |
| GPT-4o-mini | 0.8 | 1.2 | 1.1 | 0.3 | 1.087 | 1.091 | 1.097 | 1.075 |
| Sonnet-3.5 | 1.3 | 2.9 | 0.7 | 0.6 | 1.087 | 1.168 | 1.056 | 1.054 |
| Haiku-3.5 | 1.6 | 2.4 | 1.5 | 1.2 | 1.135 | 1.200 | 1.119 | 1.102 |
| Qwen2.5-vl-72b | 0.2 | 0.0 | 0.4 | 0.3 | 1.050 | 1.101 | 1.037 | 1.024 |
| Qwen2.5-72b | 1.2 | 2.5 | 0.8 | 0.6 | 1.157 | 1.273 | 1.140 | 1.085 |
| AgentOccam | 4.8 | 11.3 | 3.0 | 1.8 | 1.658 | 2.121 | 1.417 | 1.524 |
| BrowserUse (Qwen2.5-72b) | 13.2 | 21.9 | 11.0 | 8.6 | 2.044 | 2.385 | 1.993 | 1.841 |
| BrowserUse (GPT-4o) | 12.9 | 21.9 | 7.0 | 11.2 | 2.171 | 2.490 | 1.934 | 2.133 |
| BrowserUse (GPT-5.1) | 22.2 | 38.1 | 15.1 | 14.3 | 2.144 | 2.682 | 1.916 | 2.102 |

Table 5: Success Rate (SR%) and Score (1–5) of 10 models on WebDS-live benchmark. BrowserUse (GPT-5.1) performs the best with a success rate of 22.2%, most models have below 2% Success Rate. Success rate is measured by the number of tasks evaluated as SUCCESSFUL and score is given by an LLM as a judge, as defined in section 5.1. If only the model name is specified, the model is run on the base framework given in the WebArena Base Agent.

Despite strong performance on prior benchmarks, all agents perform poorly on WebDS. `AgentOccam`, for example, achieves $45.7\%$ success on WebArena but only $4.8\%$ on WebDS.

---

[3]as of 5/15/2025

| Failure Theme | Example | Analysis |
|---|---|---|
| Navigation | The model navigated to the American Physical Therapy Association instead of the American Public Transportation Association.. | The model confuses similarly named entities, interacting with irrelevant domains. It lacks validation steps to check if the information source aligns with the query. |
| UI Feedback | The model repeatedly tried to set a search field to "Abstract," but the system failed to register this action. | The model cannot confirm whether its UI manipulations are successful. This leads to broken search flows and incomplete query execution. |
| Groundedness | The model failed to mention the 12% bias in generative AI, despite viewing the correct document. | The model fails to extract key details and instead provides vague summaries. This shows a gap between information access and correct synthesis. |
| Failed Repetition | The model retries the same search-with-filter after UI feedback indicates the filter did not apply, repeating for dozens of steps. | Lacks loop-breaking and state-checking heuristics; does not verify that the previous action changed page state before retrying. |
| Query Interpretation | Asked for a 30% publication growth figure but responded with general research trends. | The model misunderstands that task require certain kinds of reasoning (e.g., a specific numeric trend, not a qualitative description). |
| Effort Allocation | Couldn't retrieve release count for an artist via MusicBrainz, then gave incorrect Wikipedia total. | The model attempts to use a tool, but after failing, simply googled the task and got it wrong because Wikipedia records *unique* releases rather than *all* releases. |

Table 6: Taxonomy of model failure themes with representative examples and underlying causes.

Likewise, `BrowserUse + GPT-4o`, which surpasses OpenAI's Operator and Claude Computer Use with an 89% score on WebVoyager, drops to just 12.9% on WebDS (even with our adaptations; see Section 8). This sharp decline underscores that web-based data science poses fundamentally different challenges from those captured by existing benchmarks.

Interestingly, BrowserUse with Qwen2.5-72B outperforms the same setup with GPT-4o, suggesting that raw model capability is not the primary bottleneck. Instead, limitations arise in the translation layer between reasoning and environment interaction, as evidenced by frequent UI handling failures in our error analysis. These findings indicate that progress may depend more on improving control and interaction fidelity than on scaling model capacity alone.

Humans, by contrast, achieve an average success rate of 90%, establishing a human–agent gap of over 75 percentage points. Human participants were able to flexibly adapt strategies, validate intermediate outputs, and recover from interface quirks, which were behaviors current agents lack. This contrast confirms both the solvability and realism of WebDS tasks while exposing critical limitations of today's agents on long-horizon, end-to-end workflows.

## 7 ANALYSIS

**Failure Analysis** We evaluated the reasons behind the failure of these LLM agents, and looked for key reasons behind their failure. Most failures stemmed from weak grounding, poor feedback handling, and misinterpretation of user intent, more details can be found in Table 6.

**Benchmark Longevity** We also design WebDS to stay relevant and avoid saturation by (i) covering diverse domains, data types, and three difficulty tiers, (ii) implementing a dual evaluation regime, including a fixed dockerized subset enabling reproducible comparisons across models and future research even when real websites change over time, (iii) offering an extensible task-creation workflow that keeps the benchmark "evergreen," and (iv) remaining hard enough that humans still outperform the best current models, leaving ample headroom for progress. See Appendix F for full details.

## 8 CONCLUSION

WebDS introduces a novel benchmark that rigorously evaluates AI agents on complex, web-based data science tasks. By encompassing structured and unstructured data analysis, multihop reasoning, tool integration, and action-based workflows, it fills a critical gap in current web-based agent evaluation frameworks. Our preliminary results reveal significant performance shortfalls in SOTA agents performance on our benchmark compared to current web agent and data analysis benchmarks. We also show a large gap between current agents ($\leq 13\%$ success) and a human baseline (90%), quantifying how far we are from reliable end-to-end autonomy on the web. We hope that WebDS provides a robust target for closing this gap and advancing practically useful web-based data-science agents. Future work in this area would include improving task diversity to encompass more action-based tasks and expansion to OS-level tasks or other enterprise workflows.

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

APPENDIX

## A    DATASET STATISTICS, MORE GRAPHS, AND FURTHER DETAILS

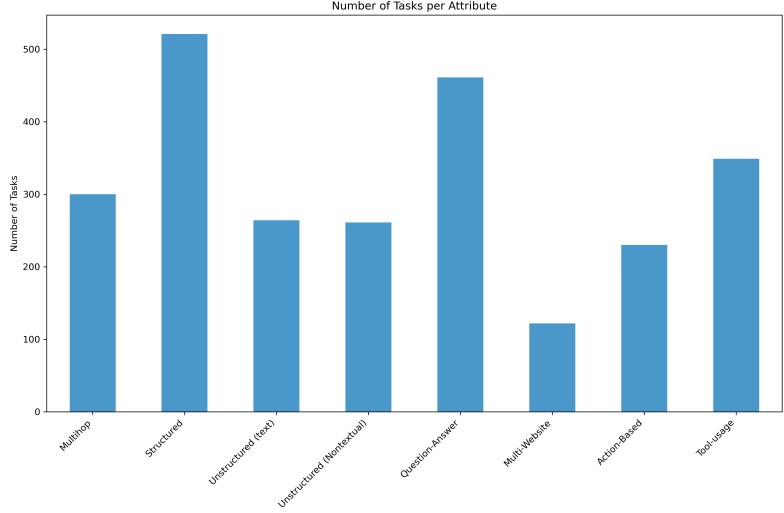

Figure 4: Counts of tasks per attribute

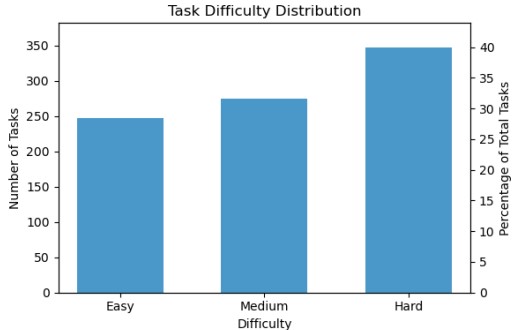

Figure 5: Counts of Task Difficulty

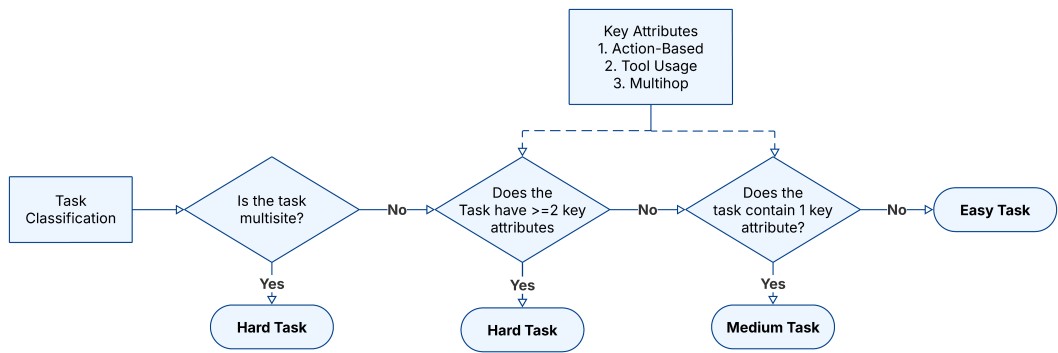

Figure 6: Flow chart for evaluating task difficulty

| Domain | Websites |
|---|---|
| Economics/Markets | BEA, FRED, ST LOUIS FED, TRADING ECONOMICS, DATAUSA |
| Demographics | WORLDPOP, WORLDOMETER |
| Music | TUNEBAT, MUSICBRAINZ, RIAA, RICHARD POWERS |
| Tourism, Trade, and Airlines | UNWTO, IATA |
| Higher Education | MIT, UCHICAGO |
| Scientific Research | ARXIV |
| Sports | UNDERSTAT |
| Government and Public Policy | CFPB, OUR WORLD IN DATA, REDDIT |
| Energy and Climate | CLIMATE GOV, NOAA |
| Health | CDC MENTAL HEALTH, CDC COVID, CDC OBESITY, NIH |
| E-Commerce | SHOPPING, STOCKNEAR |

Table 7: Mapping from Domains to Websites. **We note that the Reddit website and Shopping website are the same as in WebArena, but our tasks are unique.**

## B    SCORES FOR SPECIFIC DOMAINS

| Domain | GPT-4o | GPT-4o-mini | Sonnet-3.5 | Haiku-3.5 | AgentOccam | Qwen2.5-72b | Browser Qwen2.5 | Browser GPT-4o |
|---|---|---|---|---|---|---|---|---|
| Demographics & Policy | 0.0 | 0.4 | 0.4 | 1.2 | 2.8 | 0.8 | 10.5 | 8.2 |
| E-Commerce & Forums | 2.2 | 0.0 | 0.0 | 2.2 | 2.2 | 2.2 | 2.2 | 13.3 |
| Economics/Markets | 1.0 | 0.0 | 3.0 | 3.0 | 8.0 | 3.0 | 21.8 | 20.8 |
| Energy & Climate | 0.0 | 0.0 | 0.0 | 0.0 | 20.7 | 0.0 | 27.6 | 34.5 |
| Health | 2.5 | 3.7 | 1.2 | 3.7 | 8.6 | 2.5 | 18.5 | 16.0 |
| Higher Education | 0.0 | 0.0 | 0.0 | 0.0 | 0.0 | 0.0 | 8.0 | 4.0 |
| Music | 1.0 | 0.0 | 1.0 | 1.0 | 1.0 | 0.0 | 8.7 | 6.7 |
| Scientific Research | 1.8 | 2.7 | 2.7 | 1.8 | 2.9 | 0.9 | 12.4 | 12.4 |
| Sports | 0.0 | 0.0 | 0.0 | 0.0 | 0.0 | 0.0 | 17.6 | 17.6 |
| Tourism, Trade, Airlines | 1.2 | 0.0 | 2.4 | 1.2 | 9.6 | 1.3 | 14.5 | 18.1 |

| Attribute | GPT-4o | GPT-4o-mini | Sonnet-3.5 | Haiku-3.5 | AgentOccam | Qwen2.5-72b | Browser Qwen2.5 | Browser GPT-4o |
|---|---|---|---|---|---|---|---|---|
| Multihop | 0.7 | 0.3 | 0.7 | 1.4 | 3.1 | 0.7 | 9.5 | 11.6 |
| Structured | 0.6 | 0.8 | 0.6 | 1.0 | 3.2 | 1.0 | 11.8 | 9.4 |
| Unstructured (text) | 0.8 | 0.4 | 1.9 | 1.2 | 7.0 | 1.6 | 15.8 | 18.1 |
| Unstructured (nontext) | 1.2 | 1.2 | 0.4 | 2.0 | 1.6 | 1.2 | 8.9 | 10.1 |
| Question-Answer | 1.1 | 0.7 | 1.8 | 2.0 | 6.1 | 1.3 | 15.8 | 13.7 |
| Multi-Website | 1.7 | 0.0 | 1.7 | 0.8 | 2.5 | 0.9 | 5.8 | 6.6 |
| Action-Based | 0.9 | 0.0 | 0.0 | 0.9 | 1.8 | 0.5 | 7.9 | 13.2 |
| Tool-usage | 0.3 | 0.3 | 0.9 | 0.6 | 1.5 | 0.3 | 8.3 | 8.0 |

Table 8: WebDS benchmark performance: domain-wise and attribute-wise success rates (%).

## C    BROWSERUSE AGENT IMPLEMENTATION

We build on the open-source **BrowserUse** framework (snapshot: April 22nd, 2026) and introduce several modifications to increase robustness on our benchmark. This appendix summarizes the final agent configuration used in all experiments.

**1. Task Initialization and Control Loop**    Each evaluation episode is defined by a JSON configuration specifying a *starting URL* and *natural-language intent*. This intent is injected directly into the system prompt as the agent's "ultimate task," and the model must terminate using a `done` action once this task is completed. At every step, the agent receives the browser state (URL, rendered content, DOM/AXTree structure, screenshots) and must respond using a structured `AgentBrain` object containing:

1. an assessment of whether the previous action achieved its goal,
2. a short memory field tracking progress, and
3. the next immediate sub-goal.

This structure enforces continual micro-planning and self-evaluation, even without an external planning module.

**2. Action Space**    The agent interacts with the environment through high-level semantic actions exposed by BrowserUse, including clicking indexed elements, scrolling, typing, issuing search queries, switching tabs, and extracting text or structured content. We expand the system prompt to list *all* available actions explicitly, as we observed that the default prompting often caused the model to overlook useful but less frequently used actions.

**3. Enhanced Observation Space via AXTree Indexing**    The default BrowserUse implementation relies on a heuristic for identifying clickable elements that frequently misses important links on real webpages. To avoid navigation failures, we replace this mechanism with an **AXTree-based traversal** that assigns a unique index to every interactable anchor element. This guarantees that all visible links become addressable by the agent, substantially improving coverage and stability in multi-step navigation tasks.

**4. Lightweight Memory**    BrowserUse includes an optional procedural-memory compression module, but we disable it for our experiments. Instead, the model maintains its own high-level task-state ledger in the `memory` field (e.g., "3 of 10 pages checked"). For our task lengths, this lightweight in-context memory proved sufficient to maintain coherent long-horizon behavior.

**5. CSV Detection and Data Tools**    Many benchmark tasks require analysis of structured data presented as large CSVs on webpages. To support this efficiently, we extend the agent with:

1. automatic detection of CSV-like page content,

2. conversion of such content into local DataFrames,

3. pickle-based caching for fast re-access, and

4. dedicated actions allowing schema inspection, row extraction, and execution of safe Python analysis code.

These tools allow the agent to perform nontrivial data-analysis operations without requiring the model to ingest large tables token-by-token.

**6. Parallel Execution**    Evaluation is performed using an asynchronous multi-worker pipeline. Each worker runs an independent agent and pulls tasks from a shared queue with retry logic. When multiple API keys are available, they are distributed across workers to avoid rate limits. Full trajectories—including browser states, screenshots, and agent outputs—are logged for post-hoc analysis.

## D    COMPUTE USED

We used 4x 16-core Xeon CPUs with 256 GB of RAM between them to run agents in parallel massively. No GPUs are needed as we make direct calls to provider APIs (no locally hosted models), and running these tasks are CPU and I/O constrained.

## E    FORMALIZATION OF AGENT APPROACH

**1. Navigation and Information Gathering**    The agent interacts with the environment to extract raw data relevant to a given query:

$$f_d : \mathcal{W} \times \mathcal{Q} \to \mathcal{D}. \tag{1}$$

**Note:** In dynamic settings, $\mathcal{W}$ may be modeled as an interactive state space or a Markov Decision Process (MDP) to capture sequential interactions.

**2. Representation Learning and Outcome Prediction**    The agent processes the raw data to obtain an analytical outcome. This stage is further decomposed into:

**a. Representation Learning**    Transform the raw data $d \in \mathcal{D}$ into an intermediate representation $T$ that retains only the information necessary for prediction:

$$f_T : \mathcal{D} \to \mathcal{T}. \tag{2}$$

**b. Outcome Prediction**    Map the learned representation $T$ to an analytical outcome $y \in \mathcal{Y}$:

$$f_y : \mathcal{T} \to \mathcal{Y}. \tag{3}$$

Together, the above steps (a) and (b) define the mapping from raw data to some analytical outcome:

$$f_a : \mathcal{D} \to \mathcal{Y}. \tag{4}$$

**3. Optional Downstream Action** Based on the analytical outcome $y$, the agent may execute a downstream action:

$$f_\alpha : \mathcal{Y} \to \mathcal{A}, \tag{5}$$

with $a = \varnothing$ indicating that no further action is taken.

- **HTML** - The raw HTML structure of the web pages.
- **DOM** - The Document Object Model representation.
- **Screenshots** - Visual representations of web pages.
- **AXTree** - Accessibility tree structures for web elements.
- **Downloaded Data** - Unlike prior benchmarks, our setup allows for downloading data, which is then loaded into the LLM's context. As a result, the LLM can also observe structured data formats such as:
  - JSON
  - CSV
  - Other structured data formats.

## F   MORE DETAILS ON LONGEVITY

1. **Fine-Grained Difficulty and Domain Categorization.** WEBDS evaluates agents across multiple axes, including task difficulty, data representation (structured, unstructured), and domain (e.g., demography, finance, public health). This multi-dimensional evaluation reduces the risk of models overfitting to a narrow set of tasks and provides continued granular insight even as models improve over time.

2. **Containerized and Stable Evaluation.** By having a dockerized track in WebDS, we eliminate the instability associated with live web environments. This ensures that future researchers can reproduce results exactly and fairly compare new methods against older baselines, even years after initial release.

3. **Evergreen Task Format and Modifiability** Data science tasks inherently offer broad extensibility: new queries, updated data sources, and evolving analytical goals mean that researchers can continuously generate new tasks using the same framework. WEBDS supports community-driven extensions through tools that enable users to create, classify, and evaluate new tasks and domains with minimal overhead. In particular, this can be adapted to tackle many other enterprise workflows, showing huge potential as a benchmark to evaluate general AI agents. For example, this can be used to add specialized domains like scientific computing or finance if the community finds it useful, with the same Dockerization and rubric-based evaluation.

4. **High Task Complexity.** Even current frontier models perform poorly, indicating a wide margin for future progress. Additionally, human performance is non-trivial on many tasks, underscoring the benchmark's difficulty and its grounding in real-world challenges.

### F.1   TASK EXAMPLES

To further illustrate the breadth and depth of the WebDS task suite, we provide representative examples of tasks spanning data gathering, cleaning, transformation, visualization, statistical analysis, and modeling. These examples highlight how WebDS goes beyond retrieval or surface-level question answering to evaluate end-to-end data science workflows. Herein, we provide some examples of tasks. For long tasks, we occasionally provide source context in front of the task which is not shown here for brevity.

**Data Gathering, Cleaning, and Preprocessing.** Many tasks require the agent to extract heterogeneous data from dynamic web interfaces, resolve formatting inconsistencies, and organize the results into usable structured formats. Examples include:

- *"Extract the most difficult positions to fill in the March 2022 public transit workforce report and organize them into an Excel spreadsheet."*

- *"Align unemployment and COVID–19 time series for all countries with GDP per capita above a specified threshold and normalize the series for comparison."*

- *"Identify all product volumes that appear more than once after applying a multi-stage filtering pipeline."*

- *"Rank global GDP levels after cleaning and merging country-level datasets with mismatched units and missing values."*
- *"Clean long-horizon literacy or poverty datasets spanning multiple decades and resolve missing entries and inconsistent measurement units."*

These tasks require structured data extraction, multi-step cleaning, schema reconcilation, and the ability to construct stable tabular outputs.

**Visualization and Chart Construction.** A large subset of tasks evaluates an agent's ability to produce correct and contextually appropriate visualizations. Such tasks include:

- *"Visualize literacy trends in the United States and discuss long-run patterns."*
- *"Plot a multi-city line graph of transit ridership using data from major U.S. metropolitan systems."*
- *"Generate a chart summarizing MIT's undergraduate financial aid statistics and annotate notable trends."*

Solving these tasks requires the agent to retrieve numerical data, structure it appropriately, select a suitable visualization type, and execute the visualization using an external tool.

**Statistical Computation and Analysis.** WebDS also contains many tasks assessing competence with descriptive and inferential statistics, requiring nontrivial numerical computation. Examples include:

- *"Compute the variance of BPM across top-charting music tracks."*
- *"Calculate the standard deviation of average schooling hours from 1900 to 2005."*
- *"Evaluate efficiency statistics for high-income countries using multi-dimensional indicators."*
- *"Compute correlation coefficients and cross-correlations (with optimal lag) between student spending and literacy outcomes."*

Such tasks require the agent to derive correct numerical values, interpret temporal structure, and integrate multi-source data.

**Predictive Modeling and Regression.** A number of tasks in WebDS explicitly require model fitting and forecasting using linear, multivariate, or exponential models. Illustrative examples include:

- *"Model literacy rates using per-capita spending and healthcare expenditure as predictors, and assess model fit."*
- *"Fit a linear regression linking student spending and GDP from 1990 to 2020 and interpret the resulting coefficients."*
- *"Perform a regression between product weights and prices and evaluate the strength of the relationship."*
- *"Fit an exponential growth model to African tourism arrivals and forecast values for 2030, comparing the result to a linear model."*

These tasks evaluate the agent's ability to extract relevant features, apply statistical models, and interpret predictive outputs.

**Multisite Data Gathering and Cross-Domain Synthesis.** A core feature of WebDS is the requirement to gather data across multiple websites, often with distinct modalities, and to synthesize them to produce a coherent output. Examples include:

- *"Compare literacy rates between Latin America and the Caribbean and the rest of the world, compute interquartile ranges over the last fifty years for each region, and visualize the trends."*
- *"For countries with GDP per capita above 14,000 USD, compute the average maximum cross-correlation and corresponding lag between unemployment and COVID–19 cases."*
- *"Extract the most difficult positions to fill in the March 2022 public transit report, compile them in an Excel spreadsheet, and produce a relevant chart within the spreadsheet."*
- *"Develop a Python script to cluster adults by reported mentally unhealthy days using National Health Statistics Reports data."*
- *"Analyze regional tourism arrivals for 2010, 2015, and 2020; fit an exponential growth model for Africa; predict arrivals for 2030; and compare the forecast to a linear growth model."*

These examples highlight scenarios where agents must integrate multi-source information, perform multi-hop reasoning, and execute downstream analytical or programmatic actions.

**Interpretation of Technical Content.** Finally, WebDS includes tasks assessing the ability to read tables, interpret scientific figures, extract quantitative findings from academic articles, and synthesize domain knowledge, with the ability to do this across a wide variety of figures, papers, time periods etc. Examples include:

- Interpret ablation tables from a specific machine learning paper and computing relative performance differences

- Extract and compare model performance from figure-based visualizations and describe accuracy trends

- Identify patterns in ArXiv submission metadata from specific time periods and report anomalous trends

These tasks assess high-level reading comprehension, technical grounding, and cross-modal interpretation.

**Summary.** Together, these tasks demonstrate the breadth of operations WebDS is designed to evaluate, spanning raw data acquisition, multi-website synthesis, structured cleaning, statistical computation, visualization, and modeling. They reflect real-world analytical workflows that require coordinated reasoning, tool use, and multi-step execution, and therefore serve as a rigorous stress test for modern web-based data science agents.

## G EXPANDED FAILURE ANALYSIS

| Failure Mode | Count | Proportion (%) |
|---|---|---|
| Groundedness | 309 | 40.2 |
| Query Interpretation | 221 | 28.8 |
| Effort Allocation | 97 | 12.6 |
| Failed Repetition | 49 | 6.4 |
| Other | 42 | 5.5 |
| Navigation | 34 | 4.4 |
| UI Feedback | 16 | 2.1 |

Table 9: Distribution of annotated failure modes across 764 failed trajectories (not all failures are from the same model).

In Section 7, we introduced a high-level taxonomy of agent failures on WebDS. Here, we provide a more detailed appendix-style analysis of these error modes and their relative prevalence, based on manual annotation of 764 failed trajectories.[4] We group errors into seven themes, aligning with the qualitative examples in Table 6.

**Groundedness (40.2%)** The most common failure mode involves incorrect use of information that is actually available to the agent at *some point* along its trajectory. In these cases, the agent successfully reaches the relevant page or document but either: (i) mis-reads key numerical or categorical details, (ii) omits crucial values (e.g., a reported 12% bias figure), (iii) hallucinates facts not supported by the source (iv) uses information from a similar but not correct document or dataset, or (v) does not ground final action in an analysis of the data. Typical instances include reporting incorrect statistics from graphs; summarizing a policy document while attributing it additional constraints not present in the text; conflating similar quantities (e.g., total vs. unique counts); or not correctly using data from a document it had visited earlier. These errors indicate that the bottleneck is not necessarily just access to information, but also that agents often fail to ground answers, especially when dealing with dense tables, multi-column CSVs, or long reports.

**Query Interpretation (28.8%)** The second-most prevalent category arises when the agent fundamentally misinterprets the user's intent, even if its subsequent reasoning is internally consistent. Common patterns include:

- Responding with qualitative trend descriptions when the task requires a specific numerical quantity (e.g., a 30% growth figure).

---

[4]Counts: Groundedness (309, 40.2%), Query Interpretation (221, 28.8%), Effort Allocation (97, 12.6%), Failed Repetition (49, 6.4%), Other (42, 5.5%), Navigation (34, 4.4%), UI Feedback (16, 2.1%).

- Treating meta-requests ("extract all links," "compare these two releases") as open-ended explanation tasks.

- Answering only a subset of a multi-part query or focusing on the wrong entity or span of time.

- Provides an action where an analysis is required, or provides an analysis where an action is required.

These cases reveal limitations in intent parsing and constraint tracking. The agents often latch onto a salient fragment of the prompt while ignoring explicit requirements about outputs, granularity, or evaluation criteria.

**Effort Allocation (12.6%)**  Effort allocation failures capture mismatches between task complexity and the level of reasoning or exploration the agent performs. At one extreme, agents produce overly terse responses to multi-step analytical tasks (e.g., inspecting only the first table they encounter and drawing conclusions from partial evidence). At the other extreme, they sometimes over-elaborate on simple tasks, introducing unnecessary speculation instead of extracting a small set of requested fields. A recurring pattern is early abandonment of difficult subproblems: after encountering friction with a primary data source (e.g., a structured API-like site), the agent shortcuts to a secondary, less reliable source (e.g., a general search or a more approximate summary page), leading to incorrect results. This suggests that current agents lack robust strategies for budgeting search, analysis, and verification effort over long-horizon tasks.

**Failed Repetition (6.4%)**  Failed repetition errors occur when agents get stuck in behavioral loops or repeatedly apply a strategy that has demonstrably failed. Examples include:

- Issuing the same search-with-filter action multiple times after UI feedback has already indicated that the filter did not apply.

- Revisiting the same page and re-reading the same content without updating internal hypotheses or trying alternative navigation paths.

- Scrolling without trying another action when a page doesn't load

These failures highlight weak loop-breaking heuristics and poor use of negative feedback. The agents rarely maintain explicit "this didn't work" state and therefore lack mechanisms to systematically diversify strategies.

**Navigation (4.4%)**  Navigation failures are defined by "reaching the wrong entity, dataset, or website section, even when relevant resources are present within the environment". For example, an agent might navigate to the American Physical Therapy Association instead of the American Public Transportation Association when resolving an acronym, or select an unrelated subsection of a site instead of the specified dataset. This reflects poor ability to disambiguate and over similar datasets and sites. Although less frequent overall, they are particularly problematic in multi-site and multihop settings.

**UI Feedback (2.1%)**  UI-feedback issues arise when agents fail to correctly infer whether an attempted interaction (e.g., setting a filter to "Abstract") has succeeded. In these cases, the agent continues planning subsequent steps under the assumption that the interface is in a desired state, even when the page shows no change. This mismatch leads to broken search flows, empty result sets, or repeated no-op actions. While less common, it still indicates that robust web agents must reason not only over high-level goals, but also over the state transitions induced by actions, including recognizing when an interaction has no effect. Future work may benefit heavily from a robust way to predicting the general effect of an action.

**Other (5.5%)**  The remaining cases fall into a heterogeneous "Other" category. These typically involve minor formatting issues, incomplete outputs that are not clearly attributable to one specific category, or edge cases where multiple error types co-occur without a dominant theme.

**Aggregate Takeaways**  Two conclusions follow from this distribution. First, the dominant failure modes—Groundedness and Query Interpretation—are fundamentally about *long-horizon control and alignment*; in particular, reasoning with available evidence and user intent, over a long sequence of decisions and history of relevant data. That is, agents often reach the right place but fail to do right thing. Second, the remaining modes (Effort Allocation, Failed Repetition, Navigation, UI Feedback) emphasize limitations in short-horizon control — agents *perhaps* struggle to manage search effort, adapt to failed strategies, and maintain a consistent internal view of UI state.

## H   DOCKER IMAGE SAFETY

We scrape a large portion of our sites from the internet, to avoid unsafe images we picked websites from safe domains such as governmental organizations, nonprofits, and reputable companies with no ads.

## LLM ACKNOWLEDGEMENT

As per ICLR 2026 policy, we used LLMs in this paper to aid or polish writing, including in summarizing long sentences and paragraphs as well as proofreading and fixing grammatical errors. Additionally, this is a paper on LLMs, so LLMs were used as part of the experimentation process on the benchmark as required.

