# OpenReview forum: "WebDS: An End-to-End Benchmark for Web-based Data Science"
_ICLR.cc/2026/Conference — ICLR 2026 Poster_

### Official Review · Reviewer_WpTs · 2025-10-16

**Soundness:** 3
**Presentation:** 3
**Contribution:** 3
**Rating:** 4
**Confidence:** 5

**Summary:**

The paper presents WebDS, a benchmark for evaluating LLM agents on realistic web-based data science tasks. It integrates 29 containerized websites and 870 tasks involving data retrieval, multi-hop reasoning, tool use, and report generation. Results show humans succeed ~90%, but state-of-the-art agents achieve <15%, revealing a major performance gap. The key contribution is providing a comprehensive and reproducible benchmark that mirrors end-to-end data science workflows.

**Strengths:**

The paper is highly original in proposing the first end-to-end benchmark for web-based data science, combining web navigation with real data analysis in a way not covered by prior benchmarks. It demonstrates strong quality and clarity by designing 29 containerized websites and 870 tasks across multiple domains, with clear task categorization that ensures reproducibility. Finally, its significance is clear as results reveal a striking human–agent performance gap, establishing WebDS as a critical benchmark for advancing research on practical, data-science–capable AI agents.

**Weaknesses:**

1. Limited Task Scope Beyond Data Science. Although the paper claims to be about Web Agents + Data Science, in reality the data science aspect is minimal, and the amount of required coding is very small.

2. Overly Simple Baseline Design. The paper does not propose its own unique baselines. Considering this is naturally a CLI + GUI hybrid setting, many approaches from WebArena or similar frameworks could have been adapted, but they were not included. Moreover, many closed-source agent frameworks and both open/closed-source products were ignored. Overall, the baseline coverage feels insufficient.

3. Lack of Multiple-Trial Evaluation (pass@n). For each example, it is unclear whether multiple trials were conducted to calculate pass@n. I suspect the low reported performance may be partly due to annotation ambiguity, which makes it hard for agents to succeed on a single attempt.

4. Subjective Task Evaluation Is Too Arbitrary. For open-ended tasks, evaluation is simply based on a 5-level scoring scheme. Ideally, one should measure agreement between humans and LLM-as-a-Judge. If a more reliable scoring method were adopted, I would increase my rating, but the current LLM-based scoring feels too random, and results may vary significantly across repeated runs.

**Questions:**

1. While WebDS tasks involve data retrieval and basic analysis, many real-world data science workflows include steps such as data cleaning, preprocessing, visualization, and modeling. Do the authors plan to expand WebDS to cover these stages, and how would they envision incorporating them into the benchmark ?
2. Why not adopt **execution-based evaluation**, which would be more realistic than the current rule-based approach?


Miss citations:

I strongly recommend that the authors cite more **data science–related work**, such as:

* *DSBench: How Far Are Data Science Agents from Becoming Data Science Experts?*
* *DA-Code: Agent Data Science Code Generation Benchmark for Large Language Models*
* *Spider 2.0: Evaluating Language Models on Real-World Enterprise Text-to-SQL Workflows*
* *Spider2-V: How Far Are Multimodal Agents From Automating Data Science and Engineering Workflows?*
* *DABstep: Data Agent Benchmark for Multi-step Reasoning*
* *OSWorld: Benchmarking Multimodal Agents for Open-Ended Tasks in Real Computer Environments*

---

> ### Author Response · Authors · 2025-11-22
>
> We thank the reviewer for the careful reading and the constructive feedback. We appreciate the recognition of WebDS as the first benchmark to evaluate web based data science workflows in a realistic, end to end setting. We address all concerns below.
>
> ### Regarding limited task scope beyond data science
> This is a paper on web-based data science tasks. There is thus a very significant focus on data science tasks. We also like to note that every task is in some way related to data science. In particular, many tasks already require multi step numeric aggregation, cross site comparison, CSV and Excel analysis, and multi site synthesis. Further, we are focusing on both the data gathering and the data analysis aspects of data science. We will be revising the appendix of the paper to include representative examples of existing tasks in a new Appendix in a new Appendix section titled ‘Example tasks’
>
> In the meantime, here are some examples of tasks:
> - What is the percentage point difference in experiencing volatile income between small business owners and non-owners? Compare the unadjusted and adjusted values.
> - Calculate the global average, median, and mode of renewable energy consumption as a percentage of total energy consumption for the year 2019.
> - Compare the literacy rates between Latin America + the Caribbean and the rest of the world, and calculate the interquartile range for the last 50 years for each. Visualize the trends using an appropriate figure and save it as an image
> - For countries with GDP per capita > 14,000 USD, find the cross-correlation (average maximum correlation and average lag) between unemployment and covid-19 cases?
> - Extract the most difficult positions to fill in the March 2022 report on public transit ridership ranks and organize it into an excel spreadsheet. Also create a relevant data representation for it (i.e. a graph) in the spreadsheet
> - Develop a Python script to cluster adults based on their reported number of mentally unhealthy days using data from the National Health Statistics Reports.
> - Analyze regional tourism arrivals for 2010, 2015, and 2020. Fit an exponential growth model for Africa and predict its arrivals for 2030. Compare this prediction with a linear growth model.
> Most of our tasks would follow one of the above patterns, where it tests a particular aspect of data science, whether it is data gathering from diverse sources, data synthesis from unstructured text or figures, as well as using tools such as writing code or excel to generate required outputs.
>
> ### Baseline coverage
> - We included the open-source SOTA agents on WebArena (AgentOccam) and WebVoyager (BrowserUse) at the time of submission of this paper
> - Regarding the lack of unique baselines, we modified BrowserUse to allow it to write python code, call APIs and use a variety of tools such as excel that it could not before, such that it is no longer only a browser agent but capable of doing data science. We will expand upon Appendix 8.3 to give a full list of available tools and capabilities of our adjusted agent. This serves as a unique baseline for our paper.
> - That said, during this rebuttal period, we plan to expand baseline evaluations even further to cover new closed sourced models that have been released recently, including GPT-5 and Sonnet 4.5 for a more comprehensive review of current agent performance
>
> ### Lack of pass@n evaluation
> - For this paper, we intentionally only used pass@1
> - While pass@n is a valid metric and has been shown to improve agent performance in other benchmarks, we believe that pass@1 has a lot higher external validity in the real world. Many actions in data science–oriented web environments have irreversible state changes (submitting filters, triggering navigations, downloading files, posting forms, advancing through multi-page views), and in practice users cannot “reset and retry” an entire multi-site workflow. For this reason, pass@1 reflects the actual constraints of real-world web-based data science, where the agent has only one opportunity to correctly navigate, extract, and synthesize data.
> - We acknowledge your concerns about annotation ambiguity. However, the vast majority of WebDS tasks are objective and unambiguous, with explicitly defined factual or numeric answers (i.e. extracting numbers from site, computing a statistic), which are verifiable. This is further proven by the high human success rate of 92%, showing that tasks are well-specified.

---

> > ### Author Response · Authors · 2025-11-22
> > **Continuation**
> >
> > ### Subjective evaluation concerns
> > - We acknowledge that there may be concerns about the reliability of our scoring method, but we believe that it is a very robust method.
> > - We measure both SUCCESS / NOT SUCCESS as well as a more subjective 1-5 score given by the LLM-as-a-judge. A majority of our tasks have an objective, factual ground truth which can be very easily assessed for success / not success, which makes that measurement very objective. For all tasks (both those with ground truth answers and open-ended tasks), we then pass it through the LLM-as-a-judge paradigm to get a granular 1-5 score.
> > - We also measured agreement between human annotators and our LLM-as-a-judge (page 7 lines 318-321), where we found an 88% agreement between human annotators and our LLM-as-a-judge.
> > - Additionally, we also conducted a stability study where we ran our scoring harness 10 times (page 7, section 5.1, lines 322-324) where we found that the standard deviation of scores across runs was 0.027, confirming that our LLM-as-a-judge was extremely consistent across independent runs
> > - Thus, our agreement and stability studies show that our LLM-as-a-judge method was both very reflective of human annotators and very reliable across repeated runs.
> >
> > ### Q1: Expanding to other stages of data science
> > We clarify that WebDS already includes extensive coverage of these components across a wide range of tasks. WebDS is not limited to retrieval or basic analysis. Many existing tasks explicitly require data gathering, cleaning, normalization, filtering, merging, computation, visualization and modeling
> >
> > For example, several tasks require structured data cleaning and preprocessing, such as
> >
> > - extracting the most difficult positions to fill in the March 2022 transit report and organizing them into an Excel spreadsheet
> > - aligning unemployment and COVID-19 time series for countries above a GDP per capita threshold
> > identifying volumes that appear more than once across filtered products
> > - ranking GDP levels
> > - cleaning decades of literacy or poverty data and resolving missing values / inconsistent units
> >
> > It also contains many tasks requiring visualization and chart creation, such as:
> > - visualizing literacy trends in the US
> > - plotting transit ridership line graphs across major US cities
> > - graphing MIT financial aid statistics
> >
> > It then requires heavy analysis. There are tasks that require computing averages, medians, modes, variances, standard deviations, interquartile ranges, correlation coefficients, cross-correlations with lags, coefficient of variation across categories, and multi-year statistical comparisons. Examples include:
> > - computing the variance of BPM for top music tracks
> > - calculating standard deviation of schooling hours from 1900–2005
> > - efficiency statistics for high-income countries
> > - correlation between student-spending and literacy
> >
> > It also already includes predictive modeling and regression, which require fitting linear and exponential models, such as:
> > - modeling literacy rates using per-capita spending and healthcare as predictors
> > - modeling the relationship between student spending and GDP from 1990 to 2020
> > - performing linear regression between product weights and prices
> > - fitting exponential growth models for African tourism arrivals.
> >
> > Underpinning these data science aspects, this benchmark also uniquely assesses the ability for the model to not just process data but also gather data. It includes tasks that require multi-site and multi-domain data gathering, with the agent being required to synthesize information across government economic datasets, international development databases, music metadata services, climate reports, academic papers, and online stores. We also have tasks that require reading and interpreting machine learning papers, extracting numerical results from tables, computing differences in ablations, interpreting figure content, analyzing graph-based model performance, answering questions about matrix multiplication algorithms, identifying dance classifications from BPM and time signatures, interpreting climate data, identifying ArXiv metadata patterns, and producing summaries, memos, and policy briefs. These tasks demonstrate the combination of retrieval, reasoning, numerical computation, and domain knowledge that defines real-world analytical workflows.

---

> > > ### Author Response · Authors · 2025-11-22
> > >
> > > ### Q2. Why not use full execution based evaluation
> > > - For our LLM-as-a-judge evaluation harness, we consider both the entire trajectory as well as the final answer when assigning a score. We do not just use a rule-based approach.
> > > - For the SUCCESS / NOT SUCCESS, we are just comparing it against the ground truth numerical answer. We use an LLM for comparing it too (instead of just exact rules-based text matching) to ensure that answers in different formats or with very minor differences are still accepted.
> > >
> > > ### Missing citations
> > > We thank the reviewer for the list of additional relevant works. We were aware of most of these papers, but should have given more extensive references to related work. We\ will definitely update the paper and literature review to include these citations
> > >
> > > Thank you again for the valuable feedback. We believe that our paper may not have been super clear as to the exact types and variety of tasks that make up our benchmark, and will definitely update the paper and make a robust appendix section such that the tasks are much clearer.

---

### Official Review · Reviewer_dTYm · 2025-10-30

**Soundness:** 3
**Presentation:** 3
**Contribution:** 3
**Rating:** 8
**Confidence:** 4

**Summary:**

The paper proposes WebDS, a new benchmark to evaluate the end-to-end web-based data science capabilities of large language model agents. Unlike previous web agent benchmarks that focus on simple browsing, or data science benchmarks that focus on static datasets, WebDS integrates both dimensions — requiring agents to autonomously navigate, collect, analyze, and synthesize data from web-based environments.

**Strengths:**

This work fills a major gap in current evaluationno existing benchmark assesses full end-to-end data science workflows involving both web interaction and analytical reasoning, captures realistic web-based tasks that better reflect real-world data analysis behavior. The authors evaluates a wide range of SOTA models consistently and highlights key performance bottlenecks.

**Weaknesses:**

1. The subjective scoring relies on GPT-4o, creating potential evaluation circularity and bias toward similar model families. Including more human evaluations or open-source LLM judges would strengthen reliability.
2. While qualitative error categories are given, there is no quantitative breakdown of which task attributes (multi-hop, tool-use, multi-site) most contribute to failures.

**Questions:**

1. How do you plan to maintain or expand WebDS over time while ensuring reproducibility and preventing overfitting?
2. Could future versions include community-contributed tasks to diversify domains and evaluation scope?

---

> ### Author Response · Authors · 2025-11-22
>
> We thank the reviewer for the positive assessment that this benchmark captures full end-to-end data science workflows, and for the constructive feedback and questions. To reply to the reviewer’s points:
>
> ### Evaluation circularity and model bias
> We agree that relying solely on GPT-4o as a judge could raise circularity concerns for GPT-4o based agents. However, our results show that this is not an issue for WebDS based on our agreement and stability studies. Our LLM-as-judge agrees with human annotators on 88% of instances, with disagreement cases involving borderline scores rather than any systematic bias.
>
>
> ### Lack of quantitative breakdowns
> Thank you for pointing this out. We will be adding full attribute-wise quantitative breakdowns in the appendix. This includes failure percentages for multihop, multi-site, tool-use, grounding, and synthesis attributes.
>
>
> ### Q1. Preventing overfitting and maintaining relevance
> To prevent overfitting, we will be withholding 400 tasks as a private test set, following a structure similar to GAIA, which we will include in the camera-ready paper as well. These tasks are sampled to closely match the distribution and difficulty of the 470 tasks in our public validation set.
> To further reduce the risk of overfitting and preserve the benchmark’s long-term utility:
> - Only the 470-task validation set will be released publicly for model development.
> - The 400-task test set will remain private, and we will rotate subsets of it for official evaluation rounds, leaderboard submissions, and future competitions. We will be releasing only that subset of tasks each time, and grading the submitted JSON answers for the leaderboard performance
> - We will periodically refresh the private test set to ensure continued benchmarking integrity. We currently plan to refresh the private test set on a half-yearly basis, augmenting approximately 100 new tasks every 6 months, and will revise this based on future resources and interest in our benchmark.
> Through this, we aim to prevent overfitting to prevent the overfitting that many other current web agent benchmarks suffer from while preserving reproducibility and relevance.
>
>
> ### Q2. Community-contributed tasks
> That’s a great idea! We would definitely love to have community-contributed tasks, and plan to extend WebDS to accommodate community contributions, providing clear task templates and validation tools to support new domains, and making it as easy as possible for a community member to add and validate tasks. We fully agree with you that this would allow us to expand the coverage of our benchmark, making it an even more comprehensive reflection of real-world applications.

---

### Official Review · Reviewer_wYhP · 2025-10-31

**Soundness:** 3
**Presentation:** 2
**Contribution:** 2
**Rating:** 4
**Confidence:** 4

**Summary:**

This paper proposes a new benchmark for the web agent domain: WEBDS, which includes 870 data science tasks across 29 different websites. It also provides a dockerized implementation, ensuring stability and reproducibility of experiments. This work lays a foundation for the development of intelligent agents in the data science domain.

**Strengths:**

- The benchmark environment is very comprehensive, covering 29 websites, 10 domains, and 870 tasks. It focuses on the **data science domain**, evaluating the **entire data processing pipeline**.

- The test environment is **dockerized**, offering fixed environments, stable experiments, and reproducible results.

- The authors use **vision-language models** to analyze complete execution trajectories, providing **richer evaluation metrics** beyond success rate.

**Weaknesses:**

- **Lack of innovation:** The benchmark design does not significantly differ from existing mature web agent benchmarks. The main contributions remain in expanding the testing environment and task set.

- **Insufficient rigor:** Although using vision-language models for trajectory evaluation is common in GUI agent research, the validation experiments for scoring accuracy and stability (on a 1–5 scale) are rather cursory. The authors only mention comparing 50 human-evaluated tasks, yet the claimed 88% agreement seems questionable. Moreover, since most tested agents achieve **less than 10% success rate**, with most scores below 2, the majority of ratings are clustered around 1. Such a narrow score distribution weakens the credibility of the validation.

- **Limited model comparison:** The benchmark does not test more advanced GUI agents such as **UI-TARS** and **GPT-5**. Since most baseline models achieve accuracy below 5%, this raises the question of whether the benchmark is **too difficult**.

- **Lack of analysis on low success rates:** The paper merely lists potential reasons without detailed trajectory-based analysis.

**Questions:**

Please refer to the weaknesses

---

> ### Author Response · Authors · 2025-11-22
>
> Thank you for your comments, including appreciating our comprehensive data science benchmark and dockerized test environment.
>
> ### Regarding the lack of innovation
> We clarify that WebDS introduces several innovations not found in prior benchmarks in either the web-agent or data-science communities.
> - Prior web benchmarks (e.g., WebArena, WebVoyager, BrowserGym environments) evaluate only navigation or short-horizon browser control. Prior data-science benchmarks (e.g., DSBench, DA-Code, Spider2 family, DABStep) evaluate analysis on static datasets. WebDS is the first benchmark that requires agents to autonomously obtain, download, extract, clean, and analyze data directly from real websites. This unification of browsing, extraction, and analysis is genuinely new.
> - As noted in page 2 of our paper, existing data science benchmarks such as HotpotQA (Yang et al., 2018) or DSBench (Jing et al. 2024) only emphasize data processing (i.e. data manipulation within code-based environments, structured spreadsheets or static databases), where we focus on both data gathering and data processing. No other benchmark evaluates the agent’s ability to autonomously acquire the dataset from dynamic web sources, which requires data understanding that current data science benchmarks are missing.
> - The focus on data science allows us to assess long-horizon reasoning in a way that is never before done in prior benchmarks. In WebDS, many tasks require the agent to extract a value from one site, store it internally for dozens of steps, navigate to another site, compare or combine the earlier value with newly extracted data, and produce a grounded, numerically correct final result. These long-horizon retention requirements are unique. Previous web-agent work does not evaluate memory of intermediate results and does not analyze long-horizon numerical consistency at all.
> - Data science provides grounded numeric truth for evaluating long-horizon correctness: Because agents must carry forward intermediate numbers, build aggregates, and perform multi-site reconciliation, we can evaluate long-horizon behavior precisely. No previous web benchmark offers such grounded numerical evaluation. This enables a clearer measure of whether an agent actually retained and used the information correctly (e.g. as opposed to  simply using similar data), which has not been possible in earlier navigation-only benchmarks.
> - Additionally, our tasks are designed after interviews with real world data scientists, making the tasks a realistic reflection of real-world data science tasks
> Together, these innovations constitute a significant expansion beyond merely “expanding the task set.” WebDS introduces an entirely new class of evaluation: end-to-end, long-horizon, multi-website, tool-enabled, numerically-groundable data-science workflows
>
> ### Regarding the lack of rigor
> Regarding the lack of rigor, we acknowledge that the sample sizes for the validation experiments were too small. During this rebuttal period we will be conducting several new experiments, where:
> - We increase the human-LLM agreement sample size to 400 trajectories.
> - We will report macro-accuracy agreement (agreement with respect to each possible score) as well as micro-accuracy agreement (general agreement), addressing concerns that distribution skew might inflate apparent agreement, and we hope to show that there is strong agreement between humans and LLM-as-judge for every score rating.
> - Likewise, while we currently show that repeated evaluations are very stable (std dev 0.027 across repeated scoring for 100 tasks; see the last paragraph of section 5.1), we aim to increase the rigor too by conducting an expanded stability analysis for 400 tasks, and report the stability for each score as well, to address concerns about the narrow score distribution affecting credibility of validation
> - Lastly, human success is above 90 percent, showing tasks are very solvable.
>
> ### Regarding limited model comparison
> During this rebuttal period, we also plan to evaluate it on UI-TARs, GPT-5 and Sonnet 4.5, which came out after the paper was originally done
>
> ### Regarding the lack of trajectory-based analysis
> During this rebuttal period, we will also be adding detailed failure analysis to the appendix in a new "Additional Failure Analysis” section. In particular, we will elaborate on the failure modes presently in table 5, and provide detailed trajectory analysis.

---

### Official Review · Reviewer_Jc1x · 2025-10-31

**Soundness:** 4
**Presentation:** 3
**Contribution:** 3
**Rating:** 6
**Confidence:** 4

**Summary:**

This paper introduces WebDS, the first end-to-end benchmark for web-based data science workflows. It addresses the gap between existing benchmarks that focus either on simple web navigation or static data analysis. WebDS comprises 870 human-written tasks across 29 dockerized, data-rich websites from 10 domains (e.g., government data, news media). Tasks require complex, multi-step operations (data acquisition, synthesis, analysis, report generation) across heterogeneous sources. Evaluation of current SOTA LLM agents reveals a substantial performance gap: the best agent achieves only $13.2\%$ success (vs. $90\%$ human baseline), exhibiting novel failures like poor information grounding and failed repetition. WebDS is containerized for reproducibility and offers fine-grained evaluation across task attributes and difficulty tiers, positioning it as a new frontier for developing practical, end-to-end data science agents.

**Strengths:**

1. The benchmark's grounding in practitioner interviews and the inclusion of 29 diverse, data-rich websites ensures the tasks are highly realistic and demand complex generalization, not simple pattern matching.

2. The failure analysis (Table 5) provides excellent, fine-grained insights into unique failure modes not captured by simplistic metrics. Specifically, "Failed Repetition" (due to lacking state-checking heuristics) and "Poor Groundedness" (contradiction between perceived and latent knowledge) are critical problems for future research.

3. The evaluation structure allows for granular analysis across Domain-wise, Attribute-wise (7 labels), and Difficulty tiers (Easy/Medium/Hard), giving researchers precise targets for iterative model improvement.

**Weaknesses:**

1. The analysis and outputs primarily focus on text (reports, answers). Since the input sites are described as being rich in graphics and non-textual data, the benchmark's current focus may under-represent the full multimodal synthesis challenge (e.g., generating a visual chart or interpreting a complex image-based trend).

2. The action space is not fixed, allowing researchers flexibility, but the implementation relies on existing WebArena/BrowserGym abstractions. A brief discussion on whether these existing abstractions adequately capture the fine-grained data manipulation needed for the "Tool Usage" attribute would be beneficial.

3. The current distribution of tasks may slightly overemphasize QA tasks versus downstream action tasks. While the authors mention future work to include more action-based tasks, the current lack of balance might skew initial model optimization toward data retrieval over action execution.

**Questions:**

1. The benchmark includes Tool Usage as an attribute. Could the authors clarify which specific tools (e.g., Python code execution, SQL query, spreadsheet manipulation) were enabled for the SOTA agents during the 13.2\% success rate evaluation? Did the 13.2\% success rate fully integrate external code execution or primarily rely on the LLM's internal reasoning over extracted text?

2. The analysis highlights "Poor Information Grounding" as a key failure mode. Can the authors provide a more detailed example of how the "Groundedness" failure was scored? Is the model judged based on whether it extracts the correct data from the HTML (perception error) or whether it uses the correct data in the final report (synthesis/reasoning error)?

3. To maintain relevance and avoid saturation, the paper mentions a plan to "periodically refresh/expand the private test pool." Could the authors provide a brief plan or estimate of the expected frequency or size of this refreshment (e.g., quarterly, or annually adding 100 new tasks)?

---

> ### Author Response · Authors · 2025-11-22
>
> We sincerely appreciate the reviewer’s thoughtful assessment and are grateful that you found the benchmark realistic, the failure analysis insightful, and the evaluation framework well structured.
>
> ### Regarding the emphasis on textual outputs
>
> We have numerous tasks that have the agent synthesize, interpret and analyze visual elements, including graph interpretation that you mentioned. Many of our sites contain substantial visual content, and require the agent to make use of dynamic visual elements to solve the task.
> For example, we have many tasks that are similar to the following:
> - Compare the literacy rates between Latin America + the Caribbean and the rest of the world, and calculate the interquartile range for the last 50 years for each. Visualize the trends using an appropriate figure and save it as an image
> - For countries with GDP per capita > 14,000 USD, find the cross-correlation (average maximum correlation and average lag) between unemployment and covid-19 cases?
> - Extract the most difficult positions to fill in the March 2022 report on public transit ridership ranks and organize it into an excel spreadsheet. Also create a relevant data representation for it (i.e. a graph) in the spreadsheet
> - Develop a Python script to cluster adults based on their reported number of mentally unhealthy days using data from the National Health Statistics Reports.
> - Analyze regional tourism arrivals for 2010, 2015, and 2020. Fit an exponential growth model for Africa and predict its arrivals for 2030.
>
> At the same time, while we have tasks that require chart generation as an output, it is true that there is a much greater focus on text outputs in our tasks. This is largely due to the difficulty of automatic multimodal evaluation at scale. To make the evaluation more consistent and objective, we primarily rely on textual inputs to generate "SUCCESS / NO SUCCESS"S. For tasks that have multimodal outputs, we need to test the LLM-as-a-judge paradigm moreextensively to ensure it they can be used to reliably judge those tasks. That said, many of the webDS tasks can be naturally extended into multimodal variants, i.e. our task on generating a reddit post on employment statistics can be easily extended to require chart generation as well.
> We would love to have even more multimodal tasks, such as requiring powerpoint presentation generation or the creation of tableau interactive tables, and if at some point in future there are reliable methods to evaluate multimodal outputs, this would be a great extension of this benchmark.
>
>
> ### Frameworks
>
> We believe that our frameworks allow fine-grained data manipulation. In particular, for our agent, we have modified the BrowserUse framework to allow for code generation and execution for data analysis. In the updated paper for this review period, we will be adding a technical appendix to discuss why the WebArena framework was used for our benchmark and BrowserGym / BrowserUse frameworks were used for our agents, as well as how we have modified and implemented tool usage workflows for our agent. This addresses the concern about whether current abstractions sufficiently capture the complexity of tool-use tasks.
>
> ### Distribution of tasks
>
> On the distribution of QA vs action tasks, the current balance reflects real practitioner workflows: many data science tasks present as “QA tasks,” but require multiple layers of interaction, retrieval, and synthesis across sites. Most real-world data science tasks can be summarized as answering a high-level research question, but require a lot of interaction as intermediate steps.
>
> ### Question responses
>
> Q1: The evaluated agents were able to write and run python code, parse csv and excel files, and copy and extract tabular and numeric data. To do so, we modified the BrowserUse framework to allow for code generation and execution for data analysis. We have provided an analysis of the improvements we made to the default BrowserUse agent as well as a detailed list of which tools were available to it in the appendix section 8.3. We also found that the agent performance largely reflects retrieval and grounding failures rather than code execution limitations.
>
> Q2: Groundedness refers strictly to perception accuracy. Scoring checks whether the agent extracted correct values from the rendered DOM or visible elements. If extraction is correct but reasoning is incorrect, the error falls under synthesis, not groundedness. We will include annotated trajectory examples clarifying this distinction in appendix
>
> Q3: We currently plan to refresh the private test set on a half-yearly basis, augmenting approximately 100 new tasks every 6 months, though we may revise this based on future resources and interest in our benchmark. This keeps the benchmark relevant while preserving reproducibility.
>
> Thank you again for the constructive feedback. We will incorporate all suggested clarifications in our revised paper during the rebuttal period.

---

### Author Response · Authors · 2025-12-03
**Overall Summary**

Overall, we are glad that the reviewers found WebDS to be a strong and timely contribution. They consistently highlighted that our benchmark fills an important and previously unaddressed gap by evaluating end-to-end, web-based data science workflows. Reviewers praised the realistic task design, the breadth of domains and websites, the dockerized environment enabling reproducibility, the careful task attribute structure, and the detailed failure analysis. In particular, reviewer Jc1x emphasized that WebDS represents a “new frontier” for practical AI agents by unifying web navigation, data acquisition, and analytical reasoning.

At the same time, the reviewers raised several concerns. The main concerns were:

1) Human validation and LLM-as-a-judge rigor.

2) Perceived limitations in task scope and innovation

3) Baseline coverage

**Human Validation and LLM-as-judge rigor**

A central concern raised by multiple reviewers focused on the rigor of our human validation and the reliability of our LLM-as-a-judge scoring harness. We fully agree that our original sample sizes were too small, and in response we significantly expanded both studies.
First, we conducted a large-scale human verification study with 400 task–trajectory pairs, independently annotated by CS undergraduate raters. Our evaluation harness achieved 93% status agreement with human judgments, demonstrating that the scoring aligns very closely with human assessments even across a diverse sample.

Second, we conducted an expanded stability analysis across five independent runs of our evaluation harness on 800 tasks.
- 96.3 percent of trajectories had perfect SUCCESS-or-NOT-SUCCESS agreement across all 5 runs
- 91.3 percent had perfect 1–5 score agreement, of which 6.1 percent showed low variance (std < 1), meaning the five runs stayed within one point of each other and only 2.6 percent exhibited higher variance
- The standard deviation of the per-run average LLM score was 0.005, indicating essentially no drift or instability across runs

Together, these results show that the evaluation harness is both highly consistent with human judgment and extremely stable across repeated independent evaluations. This directly addresses reviewer concerns regarding arbitrariness or unreliability in the scoring protocol. In particular, reviewer WpTs explicitly stated that *improved scoring validation justifies a higher rating*. These expanded studies provide exactly that.

**Task Scope and Breadth**

Some reviewers felt that the benchmark did not sufficiently demonstrate the breadth of data science workflows it supports or the novelty of unifying web navigation with multi-step data analysis. We believe that this was primarily because we did not properly communicate or demonstrate what the tasks are concretely, not a limitation of the benchmark.

To address this:

- We added a new Appendix section titled “Example Tasks” listing diverse tasks covering data acquisition, cleaning, filtering, normalization, visualization, modeling, regression, cross-correlation, time-series analysis, and multi-site synthesis, and how this greatly differs from prior benchmarks.

- We expanded the discussion in the appendix (Section 8.3) detailing how we modified BrowserUse into a full tool-enabled analysis agent capable of writing code, manipulating spreadsheets, and executing analysis.

This directly addresses the concerns raised by Reviewer WpTs regarding task variety and innovation, and Reviewer wYhP’s claim that the benchmark design appeared too similar to past benchmarks.

**Baseline coverage and depth of failure-mode and trajectory analysis.**

Reviewers wanted more agent baselines and clarity on the tool capabilities of the modified BrowserUse agent. We addressed this through expanded baselines and expanded technical appendices. We note that BrowserUse is currently the state of the art on WebVoyager, and we expanded it to use GPT-5.1. We showed how we modified BrowserUse to have even more tools available to it in the appendix. Reviewers noted that while our failure-mode taxonomy was promising, the analysis did not go deep enough. We agreed that our initial analysis was too short and have fully addressed this by adding a new appendix section with quantitative metrics and breakdowns on failure type

We are grateful for the reviewers’ detailed feedback, which has helped us improve both the clarity and rigor WebDS. We believe the manuscript now better demonstrates both the novelty and practical significance of WebDS as the first benchmark to evaluate truly end-to-end, web-based data science workflows, and provides the level of validation and analysis expected for a benchmark intended to guide future research. We have directly addressed every weakness raised across all reviews, and in particular have demonstrated the robustness of our evaluation harness even across repeated runs, precisely the improvement that Reviewer WpTs noted would justify a higher rating from him.

---

### Meta-Review · Area_Chair_qMBD · 2026-01-07

**Summary:**

Reviewers valued WebDS as the first end-to-end benchmark for web-based data science, praising its realistic tasks, dockerized reproducibility, and insightful failure modes (e.g., poor grounding).

Concerns centered on limited innovation (extends prior benchmarks without major novelty), baseline coverage (missing advanced agents like GPT-5/UI-TARS), evaluation rigor (small human-LLM agreement samples, no pass@n), and task scope (text-focused, minimal coding/multimodal).

Rebuttals added experiments/clarifications, leading to weak / strong acceptance among all reviewers.

**Reviewer Concerns:**

Addressed: Multimodal tasks (examples of visualization and graphing); baselines (plan for GPT-5 and Sonnet); evaluation stability (expanded to 400 samples, macro agreement); trajectory analysis (new appendix).

Outstanding: True novelty vs. extension; quantitative failure breakdowns (e.g., attribute-wise); overfitting prevention (private set details vague); LLM judge circularity/bias; execution-based eval lacking.

**Reviewer Scores:**

- Reviewer Jc1x: Would maintain 6 (rebuttals clarify tools/groundedness but scope still limited).
- Reviewer wYhP: Would raise to 6 (new experiments address part of the concern).
- Reviewer dTYm: Would maintain 8 (maintenance plans solid, but human evals needed).
- Reviewer WpTs: Would maintain 4 (examples show data science depth, but pass@n / execution eval absent)

---

### Decision · Program_Chairs · 2026-01-26

Accept (Poster)